# Ultra-Broadband Absorber with Large Angular Stability Based on Frequency Selective Surface

**DOI:** 10.3390/ma15186452

**Published:** 2022-09-16

**Authors:** Shan Zhao, Wenyu Li, Zengrui Li, Hou Shu, Kainan Qi, Hongcheng Yin

**Affiliations:** 1School of New Media, Beijing Institute of Graphic Communication, Beijing 102600, China; 2State Key Laboratory of Media Convergence and Communication, School of Information and Communication Engineering, Communication University of China, Beijing 100024, China; 3The Science and Technology on Electromagnetic Scattering Laboratory, China Aerospace Science and Technology Corporation, Beijing 100854, China

**Keywords:** ultra-broadband, large angular stability, absorber, frequency selective surface, stealth technology

## Abstract

In this paper, a low-profile, double-layer absorber with ultra-broadband absorption and large-angle stability is proposed. In order to improve the angular stability, the square ring with concave–convex deformation is designed. It can expand the current path to realize the miniaturization of the absorber, which decreases the influence of oblique incident on absorption. The equivalent circuit model provides detailed resonance and admittance analysis, showing the existence of three resonances working together to achieve broadband absorption. The simulated results illustrate that the designed unit can achieve above 80% absorption within 2.09–18.1 GHz. The angular stability is up to 50° under TE/TM polarization with a period of 0.07 λ_L_ (the wavelength of the lowest operating frequency). The 300 mm × 300 mm prototype absorbers were fabricated for demonstration, with a total thickness of 0.096 λ_L_. The measurement results are consistent with the simulated results, which shows that the designed absorber unit can achieve ultra-broadband and large-angle absorption. The performance of devices can be widely applied in infrared detection, radiation refrigeration, and stealth technology.

## 1. Introduction

Frequency selective surface (FSS) has been widely used as a hybrid radome to reduce radar scattering cross section (RCS), suppress mutual interference, and improve radiation performance in antenna systems [1,2,3,4]. Due to its unique characteristic of regulating electromagnetic waves, FSS has attracted much attention in recent years. FSS can be regarded as a spatial filter which performs band-pass or band-stop responses to incident electromagnetic waves with a periodic structure. Meanwhile, FSS with lossy film material or loaded lumped resistors are capable of absorbing the electromagnetic wave, converting the incident electromagnetic wave energy into heat energy or other forms of energy consumption [5,6,7,8]. The lossy FSS, if carefully designed, could realize reduction in wideband RCS and be flexibly applied in electromagnetic stealth technology [9,10,11,12,13]. Therefore, it has become a hot spot among scholars.

There has been a relatively long history of research on the application of FSS on the design of broadband absorbers, such as the cascaded metal lossy layers proposed by Salisbury [14], Jaumann [15], and Bucci O [16] et al., whose distances between the cascaded layers are commonly quarter wavelength. Saptarshi Ghosh [17,18] designed multilayer resistance FSS to generate two or more resonant frequency points to expand absorption bandwidth. However, the multilayer resistive FSS structure with air layers can augment the total thickness, which is inconvenient to manufacture. Simultaneously, the bandwidth cannot be maintained against the increase in the incident wave angle [19,20,21]. Improving the absorption bandwidth with large angular stability remains a challenge and is significantly required to achieve stealth.

Since the direction and frequency of incident waves are unpredictable, it is particularly important to achieve wideband large-angle stable absorption for modern operations. Most scholars have chosen to reduce the size by designing a 2.5D frequency selective structure to extend the current path [22,23,24]. An FSS based on a 2.5D complex structure was proposed in [25], in which a 2D square helix is combined with vertical vias to achieve polarization independence for TE and TM polarizations at 2.8 GHz up to 75°. In [26], by extending the length of the air hole and the pattern shape of the folded top, the unit size of the absorber is greatly reduced. Therefore, the absorber has a broadband absorption of around 2.73~7.54 GHz at a large incident angle of 45°. The authors of [27] exploited the modal interaction poles produced by lossy arrays to achieve broadband transmission, achieving a 70% absorptive rate in a wide operating band with up to a 60° incident angle, and noted that it is difficult to achieve both a wide bandwidth and high angular stability [28,29,30,31,32].

In this paper, in order to break through the limitation between wide bandwidth and large angle, a double-layer, low-profile absorber is designed. A satisfactory absorptive rate above 80% is achieved in the range of 2.09–18.1 GHz when the incident wave angle is up to 50°, and the performance is significantly better than previous papers. This paper is arranged as follows. In Section 2, a double-layer absorbing unit is proposed and analyzed in detail along with the corresponding equivalent circuit. Section 3 illustrates the experimental results to compare with the simulation results. Finally, the conclusion is presented in Section 4.

## 2. Unit Design and Equivalent Circuit Model Analysis

The overall structure of the proposed absorber unit in this paper is shown in Figure 1a, which consists of two square F4B substrates (Sub 1 and Sub 2, ε_r_ = 2.65, tanδ = 0.002) and a square metal ground plate of the same size. The period *p* of the absorber unit is set as 10 mm, and the thickness *t* of the substrate is designed to be 0.93mm. Two air layers are sandwiched between the absorber units with heights *h*_1_ and *h*_2_, respectively. Two lossy layers, named Layer 1 and Layer 2, are also printed on the upper surfaces of Sub 1 and Sub 2, respectively. In order to achieve miniaturization to improve the absorptive rate of the unit under oblique incidence, a concave–convex rectangular ring with complex deformation is designed, as shown in Figure 1b,c. Layer 1 is composed of a convex rectangular ring while Layer 2 is composed of a concave rectangular ring and a rectangular square ring. Four chip resistors are inserted into each ring for energy absorption and consumption. Notably, the designed absorber unit cell is centrosymmetric and insensitive to polarization under normal incidence.

The equivalent circuit model (ECM) is an important approach used to predict the performance of an absorber by accurately extracting its characteristics. For a simple isotropic structure, some empirical equations can be adopted to calculate its corresponding specific inductance and capacitance, such as the square patch. For a complex structure, accurate capacitance and inductance cannot be calculated. Therefore, a “fitting” tool is adopted to derive the specific values of each resistance, capacitance, and inductance in the equivalent circuit. If both the resonant frequency and the bandwidth are considered, the value of each lumped component has reference significance.

Based on the transmission line theory, the ECM of the double-layer absorber is shown in Figure 2. Y_0_ = 1/377Ω is the admittance of free space, Y_1_ = √ε_r_Y_0_ is the characteristic impedance of the substrate. Due to its simple structure and good absorption performance, both lossy layers can be regarded as a series RLC circuit composed of resistors, inductors, and capacitor connectors. The characteristic admittances of the two resistor-inserted rings in Layer 1 and Layer 2 are denoted as Y_RLC1_, Y_RLC2_, and Y_RLC3_, respectively.

The reflection coefficient Γ of the equivalent circuit diagram in Figure 2 can be calculated according to the following formula:Γ = (Y_0_ − Y_in_)/(Y_0_ + Y_in_) (1)
where Y_in_ represents the input characteristic admittance. The input admittance can be expressed as:Y_in_ = Y_RLC1_ + Y_d1_(2)

The input admittance at different positions can be expressed as:Y_dn_ = G_dn_ + jB_dn_, n = 1, 2 (3)

When the imaginary part of the input admittance is 0, resonance can be generated. It can be expressed as:B_in_ = B_d1_ + B_RLC1_ = 0(4)

The admittance of each lossy layer is:Y_RLCm_ = G_RLC*m*_ + jB_RLC*m*_ = 1/(*R*_m_ + j(*ωL*_m_ − 1/ω*C*_m_)), m = 1, 2, 3(5)

According to the above derivation, the commercial software CST Studio Suite 2018 is used for full-wave simulation, and the equivalent circuit parameters are extracted using the Advanced Design System 2021. The reflective coefficient results are shown in Figure 3. The specific model parameter values and circuit parameters are described in Table 1. It can be noticed that three resonance points are generated at 2.86 GHz, 8.22 GHz, and 17.11 GHz, respectively. Meanwhile, Figure 4 compares the real and imaginary parts of the admittances Y_d1_, Y_d2_, and Y_in_, respectively. It can be found that the imaginary part B_in_ is almost zero within the absorption bandwidth from 2.1 GHz to 18.1 GHz, while the real part G_in_ is close to Y_0_. In other words, the proposed absorber presents good matching with free space which generates multiple resonances and broadens the absorption bandwidth.

Attributed to the coupling effect between the lossy FSS, the equivalent resistance (referring to R_1_, R_2_, R_3_) in the ECM is different from the simulated resistance (referring to *R*_a_, *R*_b_, *R*_c_) shown in Table 1. However, it can be observed that the change trend of the reflection coefficients obtained by the equivalent circuit are highly coincided with the simulated results obtained by CST. Therefore, the proposed equivalent circuit and physical structure can map each other correctly. Figure 5 plots the surface current distributions at the three resonance points, respectively. At 2.86 GHz, the current is mostly concentrated in the square rectangular ring with four resistors inserted in Layer 2. At 8.22 GHz, strong absorption occurs on the four resistors of the convex rectangular ring in Layer 1. At 17.11 GHz, the current is mainly distributed in the lossy structure of the inner concave rectangle of Layer 2. This further proves that the three resonance points are caused by the series circuits of RLC2, RLC1, and RLC3, respectively.

Absorptive rate measures the ability to absorb incident waves, which can be calculated from reflection and transmission coefficients as follows:Absorption = 1 − |S_11_|^2^ − |S_21_|^2^
(6)

Among them, since the bottom layer is a metal floor without energy transmitting, the transmission coefficient is S_21_ = 0. Therefore, the absorptive rate is:Absorption = 1 − |S_11_|^2^(7)

Since the proposed absorber is completely centrosymmetric, it has the same properties in the normally incident TM and TE mode waves. In order to study the polarization dependence and oblique incidence stability of the proposed absorber, the absorptive rate of the absorber at different incidence angles under TE and TM modes was observed. As shown in Figure 6a,b, for TE polarization, it can be seen that the absorption bandwidth does not change with the polarization angle (θ varies from 0° to 50°), and the absorptive rate is greater than 0.8. For TM polarization, the absorptive rate is approximately greater than 0.9 at an oblique incident angle up to 50°, while the absorption bandwidth is somewhat reduced.

## 3. Fabricated and Measurement Results

Based on the above analysis and simulated results, we fabricated a 300 mm × 300 mm prototype for verification. The photo of the fabricated and assembled metasurface is shown in Figure 7a. It should be noted that the proposed metasurface consists of 30 × 30 units for better coinciding with periodic structures. Each unit is a three-layer structure with two air spacers stabilized by nylon posts, of which the heights are 5mm and 7mm, respectively. Sixteen holes were dug to assemble the three-layer structure, with an inside diameter of 3 mm, as shown in Layer 1 and Layer 2 in Figure 7b,c. All resistors used in the lossy layers of these absorbers are in 0402 package, the size of which is usually 0.5 mm × 1 mm. For operating convenience, a gap of 0.6 mm width was reserved for the welding resistance, exhibited in detail in Figure 7d,e. Figure 7f describes the measurement environment in detail. Two rocker arms were installed, with the horn antennas operating at 2–18 GHz, one of which is used for transmission and the other is used for reception. The prototype is placed between two rocker arms. The reflection coefficients under different oblique incidence are measured by changing the angle between the rocker arms.

The measured absorptive rate results under two polarizations at the oblique incident angle of 10°–50° are shown in Figure 8. Because there is at least an 8° angle between two rocker arms, the ideal normal incident cannot be measured accurately. According to the measured results, it can be observed that the absorptive rates are above 0.8 and within 10°–50°. The measured results are basically consistent with the simulated results. This shows that our proposed absorber can realize broadband absorption with large angular stability. Due to some inevitable manufacture and measurement errors, the measured results are slightly different from the simulated results.

A detailed comparison table including previous papers is presented in Table 2. Note that the relative profile thickness of the proposed metasurface is 0.096 λ_L_ with a unit period of 0.07 λ_L_. In order to describe the bandwidth performance versus its thickness, the figure of merit (FoM) is provided according to [33]. Note that the proposed design has a relatively large FoM (16.5) compared with previous research studies and has an outstanding achievement in terms of the wideband absorption.

## 4. Conclusions

In this paper, an ultra-broadband, double-layer absorber with large angular stability is proposed. The two lossy layers are composed of deformed concave–convex square rings with four chip resistors inserted. Since the deformed square ring extends the current path, the unit size of the absorber is greatly reduced and the angular stability can be greatly improved. The equivalent circuit model is established to describe the resonance characteristics of the lossy layer structure in detail. It illustrates that, due to the existence of multiple resonances, the designed absorber can achieve stable absorption in an ultra-wideband range. Comparing the simulated results with the measured results, the results prove that the designed absorber can achieve an absorptive rate greater than 0.8 in the 2.09–18.1 GHz range. The angular stability is up to 50° under TE/TM polarization. Due to its miniaturized structure characteristic, the proposed absorber can greatly promote the development of broadband and large-angle absorbers. The broadband absorber with a high angular stability design has a wide range of applications in scientific research and in real life, such as stealth technology, electromagnetic interference reduction, electromagnetic isolation, etc.

## Figures and Tables

**Figure 1 materials-15-06452-f001:**
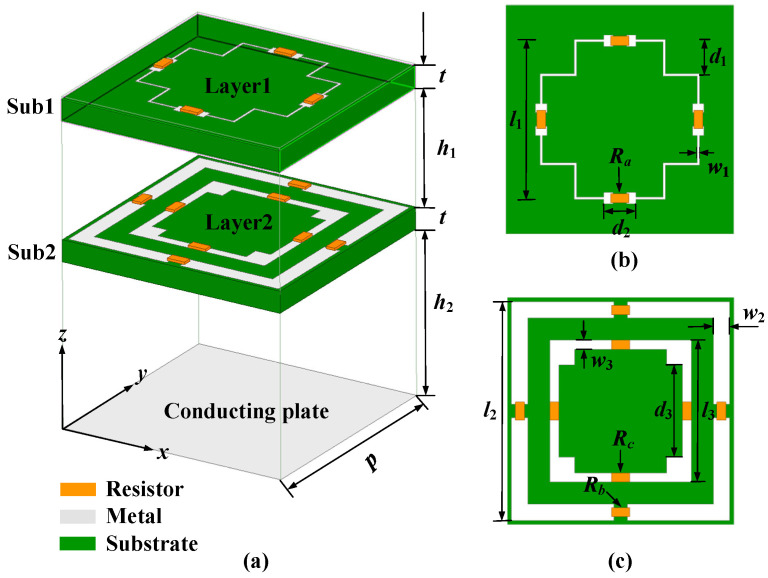
Schematic diagram of the proposed absorber unit structure. (**a**) The overall schematic diagram of the unit structure, the detailed structural schematic diagram of (**b**) Layer 1 and (**c**) Layer 2.

**Figure 2 materials-15-06452-f002:**
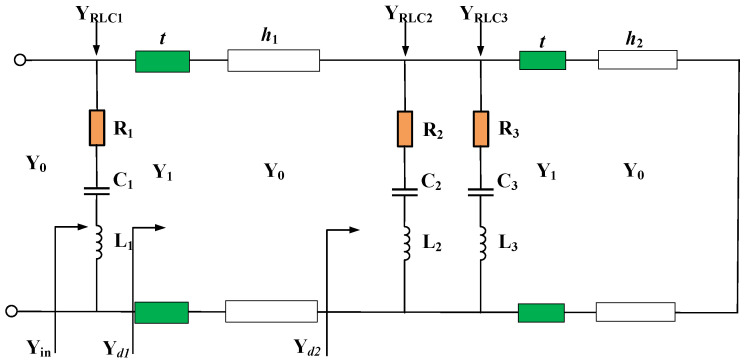
Equivalent circuit model of the proposed absorber unit.

**Figure 3 materials-15-06452-f003:**
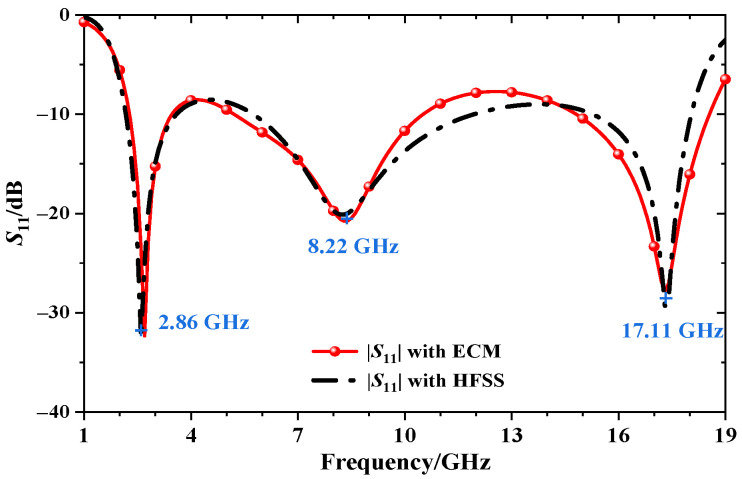
Comparison of reflective coefficients simulated by CST and calculated by ECM.

**Figure 4 materials-15-06452-f004:**
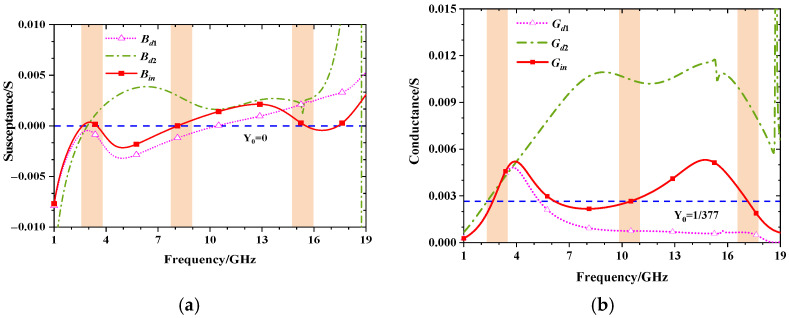
Extract imaginary (**a**) and real (**b**) parts of Y_d1_, Y_d2_, and Y_in_. The pink part represents the frequency band that meets the resonance requirements.

**Figure 5 materials-15-06452-f005:**
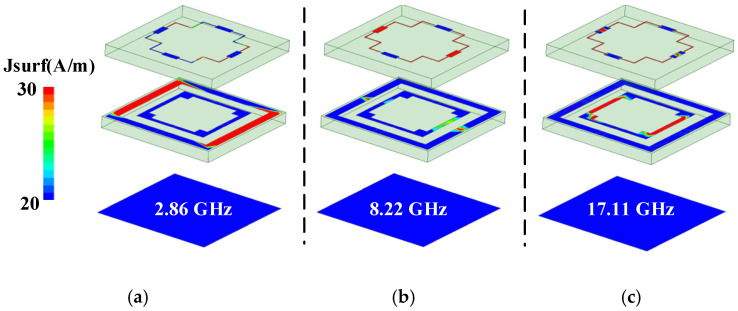
The surface current distribution at (**a**) 2.86 GHz, (**b**) 8.22 GHz, and (**c**) 17.11 GHz resonant points.

**Figure 6 materials-15-06452-f006:**
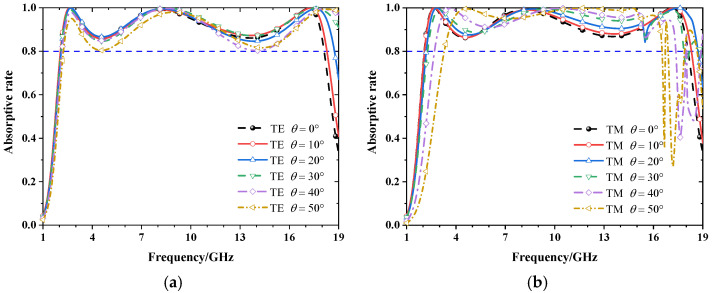
The absorptive rate at different oblique angles under (**a**) TE and (**b**) TM polarization.

**Figure 7 materials-15-06452-f007:**
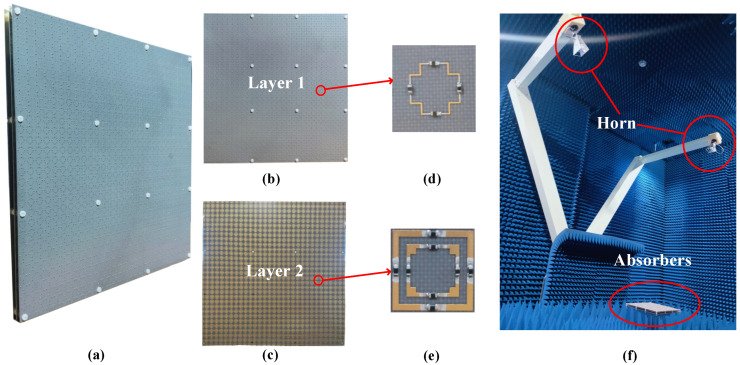
The fabricated prototype and measurement environment. (**a**) the photo of the assembled whole prototype, the front view of (**b**) Layer 1 and (**c**) Layer 2, the detailed unit photo of (**d**) Layer 1 and (**e**) Layer 2, (**f**) the measurement environment configuration.

**Figure 8 materials-15-06452-f008:**
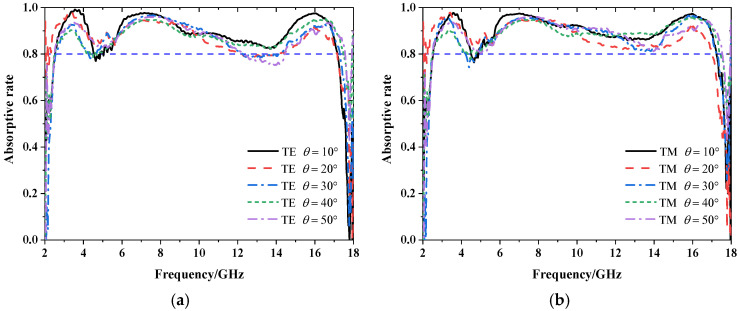
The measured absorptive rate under (**a**) TE and (**b**) TM polarizations with oblique incident angle of 10°–50°.

**Table 1 materials-15-06452-t001:** Optimized values of ECM and physical size components for absorbers.

**Parameters**	L_1_	L_2_	L_3_	**Parameters**	R_1_	R_2_	R_3_	*R_a_*	*R_b_*	*R_c_*
**Value/nH**	42.76	3.33	8.16	**Value/Ω**	353.2	198.5	233.8	400	200	100
**parameters**	C_1_	C_2_	C_3_	**parameters**	*p*	*h* _1_	*h* _2_	*l* _1_	*l* _2_	*l* _3_
**Value/pF**	0.065	0.049	0.80	**Value/mm**	10	5	7	7	9.64	6.25
**parameters**	*w* _1_	*w* _2_	*w* _3_	*t*	*d* _1_	*d* _2_	*d* _3_			
**Value/mm**	0.1	0.72	0.4	0.93	1.6	1.4	4.06			

**Table 2 materials-15-06452-t002:** Comparison of this work and previous research.

Article (Year)	Unit Thickness (λ_L_)	Unit Period (λ_L_)	10 dB RCS Reduction			Polarization	FoM	Angular Stability
			OFB (GHz)	FBW (%)	RBW (*f*_H_/*f*_L_)			
[24] 2020	0.085	0.2	2.68–12.19	127.9	4.55:1	dual	15.07	30°
[26] 2022	0.091	0.046	2.73–7.54	93.6	2.76:1	dual	10.29	45°
[27] 2019	0.12	0.16	1.6–7.1	126.4	4.44:1	dual	10.49	60°
[28] 2017	0.071	0.053	0.8–2.7	108	3.38:1	single	15.31	30°
[29] 2018	0.097	0.09	1.35–3.5	88.7	2.69:1	dual	9.12	40°
[30] 2022	0.108	0.25	1.89–6.85	113	3.62:1	dual	10.47	45°
[31] 2022	0.075	0.04	0.5–1.33	90.7	2.66:1	dual	12.09	40°
[32] 2022	0.108	0.118	2.35–6.78	97	2.89:1	dual	8.97	30°
**This work**	**0.096**	**0.07**	**2.09-18.1**	**158.6**	**8.66:1**	**dual**	**16.50**	**50^o^**

## Data Availability

Not applicable.

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
