# Peer review of "Ultra-Broadband Absorber with Large Angular Stability Based on Frequency Selective Surface"

_materials, 2022, doi:10.3390/ma15186452_

Round 1

Reviewer 1 Report

The authors have presented a low-profile double-layer absorber with ultra-broadband absorption and large-angle stability. In general, the paper is written well. I have the following observations:

1. How current path enhancement results in increasing angular stability?

2. How are the authors claiming it is a low-profile absorber when they are applying multiple layers of substrate?

3. A detailed comparison table with recently cited papers must be added in the manuscript.

4. What is the principle behind placing the chip resistors at only specific locations?

5. Did the authors consider the cross-polarization components during simulation?

Author Response

We would like to express our sincere appreciation to you for your valuable time and constructive comments. We have carefully revised the manuscript based on your comments. Please see the attachment.

Reviewer 2 Report

The authors presented Ultra-Broadband Absorber with Large Angular Stability Based on Frequency Selective Surface and the study is also presented with measured results but the topic is broadly investigated by many authors previously and I have some comments which need to be answered before it can be considered.

1. The topic presented related to broadband absorbers is already investigated by many researchers and authors do not present any comparison with these presented designs to prove that there is a certain advancement in their design compared to these designs.

2.   The writing and presentation of the manuscript can be improved. For example, equations (6-7) are blurred so need to be replaced with a word equation editor and similar mistakes need to be checked throughout the manuscript. 

3.  The figure text font in many figures is not visible. Increasing its font size will improve it.

4. In the introduction, a small paragraph related to the applications of the proposed study will improve the quality of the paper and make it more readable.

Overall, The manuscript is presenting measured results for broadband absorption but as the topic is widely investigated by many researchers, authors should provide the advancement in the study thus I am giving authors major revision so that they can correct it as per the comments given.

Author Response

(The authors gave the same response as above.)

Round 2

Reviewer 1 Report

Thank you for resolving the queries